# Stakeholder Engagement in Developing an Electronic Clinical Support Tool for Tobacco Prevention in Adolescent Primary Care

**DOI:** 10.3390/children5120170

**Published:** 2018-12-17

**Authors:** Ramzi G. Salloum, Ryan P. Theis, Lori Pbert, Matthew J. Gurka, Maribeth Porter, Diana Lee, Elizabeth A. Shenkman, Lindsay A. Thompson

**Affiliations:** 1Department of Health Outcomes and Biomedical Informatics, and Institute for Child Health Policy, College of Medicine, University of Florida, 2004 Mowry Road, Gainesville, FL 32610, USA; rtheis@ufl.edu (R.P.T.); matthewgurka@ufl.edu (M.J.G.); eshenkman@ufl.edu (E.A.S.); lathom@ufl.edu (L.A.T.); 2Division of Preventive and Behavioral Medicine, University of Massachusetts Medical School, Worcester, MA 01655, USA; Lori.Pbert@umassmed.edu; 3Department of Community Health and Family Medicine, College of Medicine, University of Florida, 1600 SW Archer Road, Gainesville, FL 32610, USA; maribethporter@ufl.edu; 4Odessa Chambliss Center for Health Equity, Bethune-Cookman University, 640 Dr. Mary McLeod Bethune Boulevard, Daytona Beach, FL 32114, USA; leed@cookman.edu; 5Department of Pediatrics, College of Medicine, University of Florida, 1699 SW 16th Avenue, Gainesville, FL 32608, USA

**Keywords:** tobacco, smoking, e-cigarettes, vaping, electronic health record, patient portal

## Abstract

Following guideline recommendations to promote tobacco prevention in adolescent primary care, we developed a patient-facing clinical support tool. The electronic tool screens patients for use and susceptibility to conventional and alternative tobacco products, and promotes patient–provider communication. The purpose of this paper is to describe the iterative stakeholder engagement process used in the development of the tool. During the pre-testing phase, we consulted with scientists, methodologists, clinicians, and Citizen Scientists. Throughout the development phase, we engaged providers from three clinics in focus groups. Usability testing was conducted via in-depth, cognitive interviewing of adolescent patients. Citizen Scientists (*n* = 7) played a critical role in the final selection of educational content and interviewer training by participating in mock-up patient interviews. Cognitive interviews with patients (*n* = 16) ensured that systems were in place for the feasibility trial and assessed ease of navigation. Focus group participants (*n* = 24) offered recommendations for integrating the tool into clinical workflow and input on acceptability and appropriateness, and anticipated barriers and facilitators for adoption and feasibility. Engaging key stakeholders to discuss implementation outcomes throughout the implementation process can improve the quality, applicability, and relevance of the research, and enhance implementation success.

## 1. Introduction

The US Preventive Services Task Force and the American Academy of Pediatrics (AAP) recommend that primary care providers (PCP) use interventions such as patient education or brief counseling to prevent tobacco use in adolescents [1,2]. Adolescents identify healthcare providers as their preferred information source regarding smoking [3,4], and parents are supportive of provider screening and counseling for tobacco use [5,6]. PCPs have unique opportunities to prevent tobacco use initiation and nicotine addiction in their patients, identify current tobacco use in both adolescents and parents, and initiate or refer for treatment [7].

Despite consensus regarding tobacco prevention recommendations, many PCPs do not routinely assess and counsel adolescents about tobacco use [8]. Reasons cited for this practice gap include lack of clear clinical practice guidelines, lack of training, and perception of low self-efficacy in delivering effective interventions [9]. Although the prevalence of cigarette smoking among adolescents has declined [10], the use of noncigarette tobacco and nicotine products is rising steeply [7], reflecting the increasing heterogeneity of tobacco product use among adolescents [11]. The probability of developing nicotine dependence and tobacco-related diseases increases with younger initiation and longer use [2]. Further, present-day adolescents who use tobacco are more likely to be from socially disadvantaged groups than in previous decades [12]. Consequently, traditional approaches to prevention may have limited fit and reach for current adolescents, given their nontraditional tobacco use patterns as well as the changing demographics of users.

Clinically efficient strategies can support providers in meeting tobacco control recommendations, but have been insufficiently implemented in adolescent primary care. Given the evolving tobacco product landscape, research needs to expand traditional strategies to address the array of tobacco products now available to adolescents. Additionally, due to the time constraints on clinical encounters, proposed interventions must achieve appropriate fit within existing clinical workflows [1,13,14]. Addressing these knowledge gaps can expand the reach of effective interventions, and improve intervention delivery, implementation, and sustainability. In response to these challenges, our team proposed developing an electronic clinical support tool to screen adolescent patients for use of or susceptibility to any tobacco product, and to promote provider-based tobacco counseling along with the delivery of brief media messages from the Food and Drug Administration (FDA)’s youth tobacco prevention campaign [15].

Interventions to improve the translation of evidence-based recommendations may substantially lower tobacco initiation and use rates. In recognition of the degree to which the engagement of multiple stakeholders prior to and during the implementation process can promote its success [16], our team engaged relevant stakeholders early in the intervention development phase. Accordingly, the purpose of this report is to describe the stakeholder engagement process used in intervention development.

## 2. Methods

### 2.1. Research Setting

The formative research described herein was approved by the University of Florida (UF) Institutional Review Board and conducted in collaboration with primary care clinics from the UF Health System, with representation from both pediatrics and family medicine. The clinics serve urban and rural communities in North Florida with large variations across clinics by income, education, and racial/ethnic minority status [17]. Additionally, UF Health is a member of the OneFlorida Clinical Research Consortium [18] and future phases of this project will involve scaling up the intervention to other health systems in the consortium. OneFlorida is a research collaborative comprising 22 hospitals, 1240 medical practices, 4000+ physician providers, and 11 million patients statewide [18]. The consortium provides a substantial research foundation with a centralized Institutional Review Board, shared governance, and implementation support from a network of practice facilitators and providers.

### 2.2. Stakeholders

We purposefully engaged various and diverse stakeholders, including scientists, clinicians, patients, and Citizen Scientists—a group of community members who fulfill the critical role of providing the patient and family perspective on clinical and translational research. Our team also collaborated with the health system clinics and information technology group. The following sections describe the Citizen Scientist program outlining the various steps in the stakeholder engagement process (Figure 1).

### 2.3. The Citizen Scientist Program

The Citizen Scientist Program is a resource of the UF Clinical and Translational Science Institute (CTSI) and OneFlorida that bridges the gap between researchers and community members. It engages patients, caregivers and other community members, who may or may not have specific health conditions, as meaningful collaborators throughout the research process. Citizen Scientists are CTSI employees who offer a lay perspective in proposal review, patient recruitment strategies, and other areas where stakeholder engagement is needed. The program includes a diverse group of participants who vary by gender, age (as young as 15), race, ethnicity, and cultural background.

### 2.4. Step 1: Implementation Science Studio

The proposed intervention was initially presented at an Implementation Science Studio organized by the UF CTSI. These studios are multidisciplinary roundtable discussions of implementation research proposals, attended by several scientists, clinicians, research coordinators, and Citizen Scientists. Participants provide concrete and actionable advice on research proposals. Although the studios support projects at any stage, they are particularly appropriate for the formative stage of development. The studio allowed us to propose our idea to a group of fellow researchers, including methodologists and clinician scientists, in addition to representatives of the Citizen Scientist Program.

### 2.5. Step 2: Consultations with Citizen Scientists

With studio discussion feedback, our research team developed the first prototype of the electronic tool and sought input on initial content of the tool through several follow-up meetings specifically with five young Citizen Scientists (3 females and 2 males, age range 15–20), who more closely resembled the target patient audience for the intervention. The Citizen Scientists were asked to critique and validate the screening questions and rank order various image- and video-based health messages from the FDA’s tobacco prevention campaign [15]. Following Proctor’s Framework for Implementation Outcomes [19], the Citizen Scientists also provided feedback on the tool’s acceptability (degree to which the intervention is “agreeable, palatable, or satisfactory” to stakeholders) and appropriateness (perceived “fit, relevance, or compatibility” for the clinical setting and problem).

### 2.6. Step 3: Provider Focus Group Discussions

We conducted provider focus groups in three pediatric clinics with 24 total participants (physicians, nurses, nurse practitioners, and office staff) to identify relevant barriers and facilitators to implementation. Guided by Proctor’s Framework for Implementation Outcomes [19], the focus groups addressed four relevant outcomes: acceptability, adoption, appropriateness, and feasibility of the intervention within the context of clinical workflow. They were conducted by trained moderators at three stages of the intervention development, whereby participants in the initial focus group were presented with a prototype of the electronic tool (developed with feedback from the studio and further refined following Citizen Scientist consultations), and the latter two focus groups tested iterations of the tool that were informed by prior focus groups. Focus groups lasted 45 min on average, and were audiorecorded for transcription and analysis.

### 2.7. Step 4: Patient Interviews

After mock-up interviews with Citizen Scientists, usability testing of the electronic tool was conducted via in-depth, cognitive interviewing of 16 adolescent patients (9 females and 7 males, 7 were 12–14 years old and 9 were 15–17 years) recruited from the UF Adolescent Clinic. This clinic mostly serves Medicaid recipients (80%) and is racially and ethnically diverse, with patients evenly divided between African American and white adolescents and roughly 10% being Hispanic adolescents [17]. The interviewed patients included 3 non-Hispanic blacks, 4 Hispanics, 3 non-Hispanic whites, and patients of mixed/other race(s). Participants were either tobacco users (*n* = 2) or determined to be susceptible to tobacco use (i.e., lack of a firm commitment to avoid tobacco use; *n* = 14) [20,21]. The following three questions were used to determine susceptibility: (1) “If one of your friends offered you a cigarette, e-cigarette, or another tobacco product, would you use it?” (2) “At any time during the next 12 months, do you think you will use cigarettes, e-cigarettes, or another tobacco product?” (3) “Do you think you will be using cigarettes, e-cigarettes, or another tobacco product five years from now?” Response options were “definitely yes”, “probably yes”, “probably not”, “definitely not”, and “don’t know”. Those who responded with “definitely not” to all three questions were considered not susceptible to cigarette smoking, whereas all other participants were considered susceptible. Adolescents were asked about their perceptions and attitudes concerning tobacco products, their personal experiences with tobacco products, and their opinions of the prototype tool. The interviews therefore ensured that all systems were in place for implementation, assessed ease of navigation by adolescents, and provided important information needed to tailor the tool to local clinical settings. Patient interviews allowed us to validate the intervention design and ensure acceptability and appropriateness among the target audience. The interviews were moderated by two researchers trained in qualitative methods, and were audio recorded for subsequent transcription and analysis.

### 2.8. Step 5: Citizen Scientist Debriefing

At the conclusion of the formative phase of the project, the investigators reconvened the Citizen Scientists to demonstrate the final product and provide them with information on how their feedback was incorporated into the study design.

### 2.9. Data Analysis

Analysis of focus group and interview transcripts was conducted concurrently with data collection to ensure that emerging themes could inform refinements to both the support tool and to the moderator guides. For both types of data, the analytic approach involved an initial phase of deductive coding (using codes derived from the research questions) and a secondary phase of inductive coding (generating new codes to assign to emerging themes) [22]. Separate codebooks for the provider focus groups and adolescent interviews were developed iteratively, and transcripts were back-coded with emerging themes identified during the inductive phase. To address inter-rater reliability, each transcript was coded independently by two trained coders, and team meetings were held to discuss and obtain consensus on coding discrepancies [23].

## 3. Results

### 3.1. Intervention Characteristics

Our team developed the tool integrating into a health risk assessment process administered via the patient portal and validated by stakeholder feedback throughout the intervention development process (Figure 2). The decision to integrate tobacco screening into the health risk assessment in the patient portal and the process of developing the tool resulted from key stakeholder recommendations. The questionnaire (Appendix A) appears in the patient’s portal inbox when scheduled for a well-child visit, and can be completed prior to or upon arrival in the clinic. Patients who are unable to enroll in the patient portal due to time constraints are offered the questionnaire via the electronic health record (EHR) system in the examination room. The tool screens adolescents for tobacco use (including cigarette smoking and use of non-cigarette tobacco products), parental tobacco use, and susceptibility to using tobacco products [20,21]. In addition, the adolescent is shown a brief video from the FDA’s campaign [15,24], for which acceptability has already been demonstrated among adolescents [15,24]. The specific video content that appears is based on patient responses about the tobacco product(s) they use and their preferences regarding the tobacco-related health message topic they are most likely to discuss with the provider. The provider views a synopsis of the questionnaire responses (including video topic) in the EHR prior to counseling the patient.

### 3.2. Implementation Science Studio and Citizen Scientist Contributions

Feedback from the studio and Citizen Scientist consultations highlighted the importance of screening for non-cigarette products, intervention with susceptible non-users, and completion time, ensuring that the tool was streamlined to optimize its fit into clinical workflow. These consultations also prepared the research team for our initial interactions with the end-users by raising critical preliminary questions and potential barriers, such as patient confidentiality and health literacy, that were included in the development of the focus group and interview guides with the end-users.

### 3.3. Focus Group Findings

Integration of the tool into clinical workflow was the most common theme in focus groups. Providers expressed concerns about “adding one more thing” to do in clinical encounters where time is already limited, and discussed ways they could adapt clinical workflow or use existing resources to accommodate the tool. As one provider noted:
“There could be potentially an office policy that says, ‘This patient is going to go into this room, fills this out.’ And have them do a survey in that area as they are checking in. That could potentially work. But, if we are doing it at the point where the nurses are already with everyone, there could be a pretty big disruption in flow.”

In the first focus group, providers were presented a version of the tool that was intended for completion on a tablet device in the waiting room (prior to the adolescent’s appointment). This format raised questions about how the tool would be administered in a way that would ensure confidentiality, as parents could easily see the screen and their child’s responses to the survey. The providers proposed solutions that could address both workflow and confidentiality issues, such as integrating the tool with intake forms already used in well-care visits, and allowing for a web-based option that could be completed on the patients’ own devices or at home prior to their visit.

This feedback led to critical enhancements to the intervention, including its integration into the patient portal platform and incorporating the AAP’s Bright Futures health risk assessment [25] (previously in paper form) into the electronic questionnaire. Building the tool within the patient portal furthered its integration with common clinic processes, streamlined patient accessibility, reduced clinical burden, and addressed concerns about confidentiality. The idea of integrating the tool with existing clinical processes, rather than trying to adapt workflow to accommodate the tool, resonated with many providers. With these modifications, the tool would not only fulfill its own function to support tobacco screening and counseling, but also introduce efficiencies to clinical workflow generally. As one provider stated:
“I think that’s the wave of the future. Why are we filling out forms? They fill out the same thing time and time again. A patient writes on the form, checks these things off, and then I take what they wrote and I type it into the computer... I’m repeating unnecessary work. Wouldn’t it be more efficient it they could type it in? It automatically goes to [the patient portal], automatically populates here every time.”

Providers and clinic staff were also supportive of utilizing the patient portal because the health system had opted to designate a version specifically for adolescents (ages 12–17 years) to maximize confidentiality in compliance with state privacy laws [26]. This innovative adaptation ensured that the implementation of meaningful use requirements aligned with the goals of adolescent health. One participant remarked that this was especially important for younger adolescents whose parents are often present during the clinical encounter:
“I truly see that as an advantage of your process… I think that adolescents are going to feel like their privacy is better protected if they can [do it] on their iPhone, their iPad, or at home prior to the clinic visit. Enter this data, but be reassured and know that it’s going to be protected and kept away from their parent.”

Furthermore, these enhancements are expected to increase acceptability and sustainability, given that increasing patient portal enrollment and implementing Bright Futures^TM^ are two initiatives with strong organizational support at UF Health. Consistent with the Diffusion of Innovation Theory [27], bundling the tool with an electronic version of Bright Futures^TM^ increases relative advantage to clinics.

### 3.4. Patient Interview Findings

Interviews with patients provided us with the opportunity to validate the intervention design and ensure acceptability and appropriateness among the target audience. By completing the questionnaire during the interview, patients confirmed initial feasibility of the tool prior to implementation in the clinic. This included an assessment of content (relevance and comprehension of the survey questions, images, and videos) and function (ease of use and navigation). Interview participants considered the intervention content to be relevant to teens their age, and favorably viewed the proposed method of completing the tool using the patient portal.

Acceptability of the tool by adolescents was high; nearly all participants considered the tool easy to use and expressed positive views about the tool. In particular, adolescents valued that the tool focused on not only the health consequences of using tobacco products, but also the social consequences that result from the effects of tobacco use on physical appearance and the cost of tobacco products. Some adolescents viewed the screener favorably because it asked about smoking by family members and friends. Two commented on questions from the susceptibility screener, stating they liked how the tool asked whether they would use tobacco if one of their friends offered it to them—a question that acknowledges social influence as an important element of susceptibility. However, another adolescent reported having trouble grasping the third susceptibility question (using tobacco “5 years from now”), remarking that it seemed “a little far off”. With regard to family, one 14-year-old participant mentioned that her father and other family members smoked cigarettes, and that because of this she had considered trying cigarettes herself. When asked what she liked about the screening questions, she noted:
“The questions were about teens, which is okay. And then we were asked about the family on the survey. I liked it because it shows how smoking could come from the family members around you. You’re seeing it everywhere you go.”

Furthermore, the views of adolescent participants on confidentiality aligned largely with the perspectives of providers. Most adolescents expressed a preference for completing the tool at home (using the patient portal) rather than in the doctor’s office, both because the portal allows for more privacy and because the home is a more comfortable setting. Although only two participants reported having ever used tobacco, most adolescents recognized that their peers who did use tobacco would be discouraged from responding to the screener honestly if measures were not taken to ensure privacy. As one 16-year-old male participant noted:
“Some teenagers, they feel like they don’t want to write so honestly. It all depends on the teenager… They’re not supposed to be doing stuff like that. They’re not supposed to be using tobacco or anything. They’re probably afraid that the person they’re talking to would go tell their parents or something.”

Although findings from the adolescent interviews did not ultimately inform substantive changes to the tool, as end-users their perspectives were critical for validating the design and usability of the tool, as well as triangulating and adding context to the perspectives of providers.

### 3.5. Debrief Findings

The Citizen Scientists welcomed the update from the research team and were pleased their contributions to science were valued. Closing the feedback loop with them was an integral component of the stakeholder engagement process.

## 4. Discussion

We deliberately engaged multiple stakeholders in the development of a clinical support tool using a combination of traditional methods (focus groups and interviews) and more novel resources (Implementation Science Studio and Citizen Scientists). The innovative tool is comprehensive in its content (e.g., screens for non-cigarette products, susceptibility, parental tobacco use, and other risk factors), and delivers tobacco prevention video messages tailored to adolescents. Following the iterative stakeholder engagement process, the tool evolved as a result of several adaptations to the initial prototype. The resulting intervention is embedded into the health risk assessment process and integrated into the EHR system and patient portal, as well as clinical workflow. All of these features are expected to increase the acceptability and sustainability of the tool in clinical practice.

Electronic health record systems offer easy-to-access regulatory-compliant methods to document assessment and counseling provided during a medical encounter with adolescent patients. Further EHR automation provides a sustainability advantage over prior interventions [28,29,30] that relied on in-clinic and post-visit staff support. Whereas EHR-based tobacco control interventions have been effective in adults [31,32,33], fewer efforts have been made to develop clinically efficient solutions for adolescents [34].

There are notable strengths and limitations to report. The key strengths of our approach include the elicitation of multiple stakeholder perspectives across the intervention development process, as well as the ability to triangulate the input received from these stakeholders using qualitative research methods. Another strength to our approach was that we preemptively discussed implementation outcomes with the various stakeholders during the intervention development phase, which is expected to maximize the intervention’s chances for success and sustainability. Conversely, this particular approach raised stakeholder concerns related to implementation that have yet to be validated in a randomized controlled trial.

The focus of the intervention on adolescents who are susceptible to tobacco use was another strength. However, our study was limited by the small number of adolescent stakeholders who reported any history of tobacco use—a critical subgroup whose perspectives may not have been sufficiently addressed in the study. The low proportion of actual tobacco users relative to susceptible individuals is typical in the adolescent population and may be misrepresented by potential underreporting of tobacco use by study participants. Nevertheless, while thematic saturation was reached overall with the 16 adolescent participants, it is unlikely that saturation was achieved with the adolescent tobacco users in our sample.

In summary, stakeholder engagement ensured that the end-users (i.e., patients and clinic staff) were represented in the development of the intervention. Citizen Scientists were also valuable partners in implementation research because they were able to identify barriers and facilitators prior to implementation. Even though the intervention is in the early stages of implementation and it is premature to evaluate the full benefits of the engagement activities, the tool has been successfully implemented in one clinic with few obstacles for over six months. Engaging key stakeholders throughout the implementation process can improve the quality, applicability, and relevance of the research, and enhance implementation success.

## Figures and Tables

**Figure 1 children-05-00170-f001:**
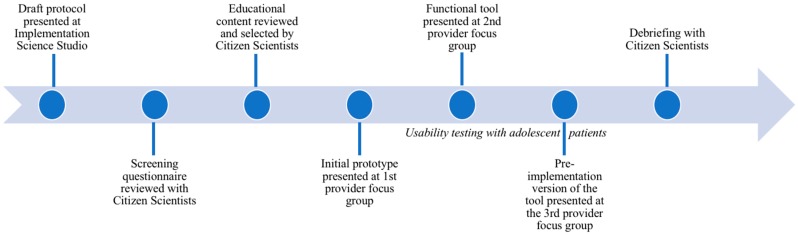
Stakeholder engagement timeline.

**Figure 2 children-05-00170-f002:**
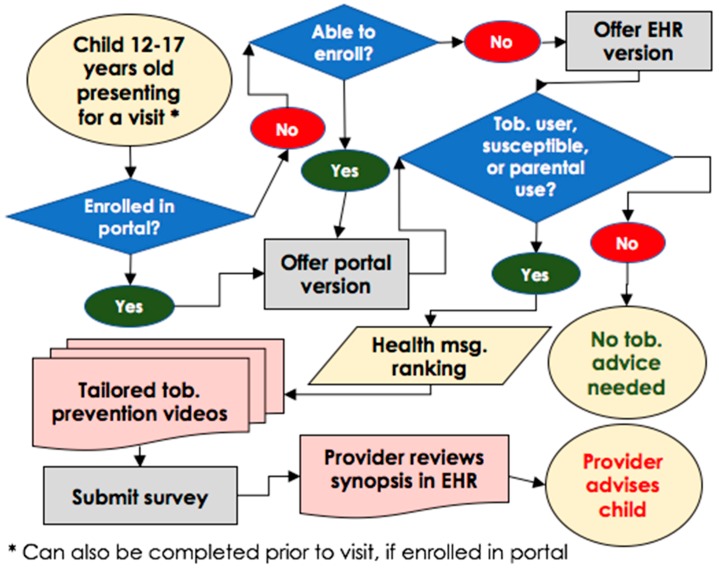
Electronic health record (EHR)-based screening clinic workflow.

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
