# Peer review of "Stakeholder Engagement in Developing an Electronic Clinical Support Tool for Tobacco Prevention in Adolescent Primary Care"

_children, 2018, doi:10.3390/children5120170_

Reviewer 1 Report

A very well designed formative qualitative study and a well-written paper.

I would like some details in this manuscript about how the susceptibility to tobacco or lack of firm commitment to avoid toacco was assessed in addition to the references provided.

The paper is about the development of a tool but we are left wondering about the tool itself - what does it include? Would it be possible to include the tool as a a supplementary file?

Please add the citation on page 3 line 119 - it simply says (cite) at the moment.

Author Response

A very well designed formative qualitative study and a well-written paper.

1. I would like some details in this manuscript about how the susceptibility to tobacco or lack of firm commitment to avoid tobacco was assessed in addition to the references provided.

Response:The questions used to assess susceptibility have been added in the manuscript (Page 4).

2. The paper is about the development of a tool but we are left wondering about the tool itself - what does it include? Would it be possible to include the tool as a supplementary file?

Response:The questionnaire has been included as a supplementary file.

3. Please add the citation on page 3 line 119 - it simply says (cite) at the moment.

Response:The citation has been added.

Reviewer 2 Report

This study aims to describe the stakeholder engagement process in developing a electronic clinical support tool to screen teens for susceptibility for any tobacco product. The discussion of a clinical tool to screen for tobacco susceptibility is highly novel. Strengths include use of Proctors Framework for Implementation outcomes use of easy to understand Figures. A few minor comments:

The Background is well written and provides justification for the study.

Methods:

Why was a young adult Citizen Scientist group used- rather than a parent and teen group?

How does the tool screen for susceptibility to tobacco products?

Results:

A highly novel aspect of the clinical support tool is that it screens teens for susceptibility to any tobacco product- Did any of the stakeholders speak specifically about the utility of the tool for screening for susceptibility to a tobacco product?

Results:

It would be helpful to have a short description of the demographics of the Citizen Scientists and the adolescents who participated in the interviews. In terms of the adolescents, would be helpful to know the n who were tobacco users or susceptible to tobacco use, as well as gender and some age breakdown.

Author Response

This study aims to describe the stakeholder engagement process in developing an electronic clinical support tool to screen teens for susceptibility for any tobacco product. The discussion of a clinical tool to screen for tobacco susceptibility is highly novel. Strengths include use of Proctors Framework for Implementation outcomes use of easy to understand Figures. A few minor comments:

The Background is well written and provides justification for the study.

Methods

1. Why was a young adult Citizen Scientist group used- rather than a parent and teen group?

Response: We worked primarily with three Citizen Scientists who constituted the “teen” Citizen Scientist group. We supplemented this group with an additional two individuals from the adult Citizen Scientist group who were in late adolescence, based on availability.  The statement in the manuscript has been revised to de-emphasize that the two represented a “young-adult” group, and rather were chosen to resemble our target population as much as possible.

2. How does the tool screen for susceptibility to tobacco products?

Response:The questions that assess susceptibility have been added. See related response to Reviewer 1.

Results

3. A highly novel aspect of the clinical support tool is that it screens teens for susceptibility to any tobacco product- Did any of the stakeholders speak specifically about the utility of the tool for screening for susceptibility to a tobacco product?

Response:We have now expanded on our findings from the adolescent patient interviews with regards to the assessment of susceptibility (Page 7, 1stparagraph). While we received overall positive feedback on acceptability from the providers, there were no specific comments on the susceptibility questions. Overall, assessment of susceptibility was determined to be important at the design stage given that greater than one-quarter of never-smoking adolescents are susceptible to tobacco use, and susceptibility predicts future tobacco use. 

4. It would be helpful to have a short description of the demographics of the Citizen Scientists and the adolescents who participated in the interviews. In terms of the adolescents, would be helpful to know the n who were tobacco users or susceptible to tobacco use, as well as gender and some age breakdown.

Response:Available characteristics have been added for Citizen Scientists (Page 3) and adolescent patients (Page 4).